# International Total Worker Health: Applicability to Agribusiness in Latin America

**DOI:** 10.3390/ijerph18052252

**Published:** 2021-02-25

**Authors:** Diana Jaramillo, Lyndsay Krisher, Natalie V. Schwatka, Liliana Tenney, Gwenith G. Fisher, Rebecca L. Clancy, Erin Shore, Claudia Asensio, Stephane Tetreau, Magda E. Castrillo, Ines Amenabar, Alex Cruz, Daniel Pilloni, Maria E. Zamora, Jaime Butler-Dawson, Miranda Dally, Lee S. Newman

**Affiliations:** 1Department of Environmental and Occupational Health, Center for Health, Work & Environment, Colorado School of Public Health, University of Colorado, Anschutz Medical Campus, 13001 E. 17th Pl., 3rd Floor, Mail Stop B119 HSC, Aurora, CO 80045, USA; lyndsay.krisher@cuanschutz.edu (L.K.); natalie.schwatka@cuanschutz.edu (N.V.S.); liliana.tenney@cuanschutz.edu (L.T.); Gwen.Fisher@colostate.edu (G.G.F.); eshore@unc.edu (E.S.); JAIME.BUTLER-DAWSON@cuanschutz.edu (J.B.-D.); MIRANDA.DALLY@cuanschutz.edu (M.D.); LEE.NEWMAN@cuanschutz.edu (L.S.N.); 2Colorado Consortium on Climate Change and Human Health, Anschutz Medical Campus, University of Colorado, 13001 E. 17th Pl., 3rd Floor, Mail Stop B119 HSC, Aurora, CO 80045, USA; 3Department of Psychology, Colorado State University, Fort Collins, CO 80523, USA; rclancy@rams.colostate.edu; 4Pantaleon Group, Guatemala City 01010, Guatemala; cdeberger@pantaleon.com (C.A.); stephane.tetreau@pantaleon.com (S.T.); ines.amenabar@pantaleon.com (I.A.); acruz@pantaleon.com (A.C.); williams.pilloni@pantaleon.com (D.P.); 5Pantaleon Group, Chinandega, Nicaragua; magda.castrillo@pantaleon.com; 6Pantaleon Group, Panuco 93990, Mexico; maria.zamora@pantaleon.com; 7Department of Medicine, School of Medicine, Division of Pulmonary Sciences and Critical Care Medicine, Anschutz Medical Campus, University of Colorado, 13001 E. 17th Pl., 3rd Floor, Mail Stop B119 HSC, Aurora, CO 80045, USA

**Keywords:** health promotion, agriculture, health climate, safety climate, health risk assessment, occupational health and safety

## Abstract

Total Worker Health^®^ (TWH) is a framework for integrating worker and workplace safety, health, and well-being, which has achieved success in European and US settings. However, the framework has not been implemented in Latin America or in agricultural sectors, leaving large and vulnerable populations underrepresented in the implementation and evaluation of these strategies to improve safety and promote health and well-being. This study presents a case study of how a TWH approach can be applied to a multinational Latin American agribusiness. We describe the process and adaptation strategy for conducting a TWH assessment at multiple organizational levels and in multiple countries. We follow this with a description of a TWH leadership training that was conducted based on the results of the assessment. Finally, we describe our methods to make corporate recommendations for TWH policies and programs that were informed by the TWH assessment and leadership trainings. With this case study we aim to demonstrate the importance and feasibility of conducting TWH in Latin America.

## 1. Introduction

The United States National Institute for Occupational Safety and Health defines Total Worker Health^®^ (TWH) as the policies, programs and practices that integrate work-related safety and health hazard protection with promotion of injury and illness prevention efforts to advance worker well-being [1]. This approach aims to integrate traditional occupational safety and health programs, broadening it to include other job-related factors (e.g., wages, benefits, workload, stress) [1]. Although recent reviews show that TWH approaches have some global reach, the vast majority of TWH intervention research has been conducted in the United States and European countries [2,3,4]. Some organizations as well as governmental entities in Latin America acknowledge the importance of this integrated approach to occupational health and safety [5,6,7,8,9], but to our knowledge there are no studies exploring the implementation of the TWH approach in Latin America, a region which encompasses many low- and middle-income countries and diverse cultures. Therefore, the benefits derived from successful implementation of TWH interventions have been limited to predominantly white populations in developed economies from western Europe and the United States. When implemented in low-income settings, such as manufacturing plants in India, it has proved to be challenging to implement such interventions [10]. Furthermore, TWH interventions have focused mostly on the manufacturing [11,12,13], construction [14,15], health services [16,17,18], and transportation sectors [19,20], while the agricultural sector has largely been absent from this research [21]. This study contributes to the literature by examining the application of TWH interventions in the agricultural sector [22] and empirically testing the cross-cultural applicability of the TWH approach.

The World Bank Organization [23] estimates that workers in Latin America comprise approximately 9% of the global labor sector, representing 317 million workers. Economic development in Latin American countries relies primarily on the mining, construction, agriculture, and fishing sectors, which are also occupational sectors with high levels of risk for work-related injuries and fatalities [24]. Indeed, the Pan American Health Organization reported that as of 2010, 40–60% of the workforce in Latin America is routinely exposed to occupational hazards, and estimated that about 11% of the global burden of occupational fatalities happen in Latin American countries [25]. The Latin American workforce is therefore both large and vulnerable. Although TWH strategies may be promising avenues to protect and promote the health of this population, they have rarely been applied to this setting.

In this paper we illustrate how to apply the TWH approach to the Latin American agriculture industry using a case study from a large multinational agribusiness conducted from 2018 to 2020. We used the case study to explore three novel concepts: the applicability of TWH theories in a Latin American cultural setting, the applicability of TWH in a vertically integrated agribusiness, and the relevance and applicability of the TWH approach within a multinational organization. Here we present our TWH conceptual model, an overview of our TWH assessment strategy, and an overview of the TWH leadership trainings aimed to improve the culture of health and safety of the organization through leadership skill development. We also present recommendations for corporate TWH policies and programs that were informed by the data collected during the TWH assessment. Due to the volume of data gathered through this assessment, this paper will focus on the methodological and conceptual approach of this work, while future manuscripts will present more detailed analysis of the data collected.

This case study represents a way to reach thousands of vulnerable workers in a large Latin American agribusiness and the opportunity to explore the applicability of TWH research, translation, and dissemination across different countries and cultures.

### Conceptual Model

Although a number of TWH conceptual models and frameworks have been developed [3,26], we based our model on work done by Schwatka et al. [27] due to its applicability for the diversity of businesses and the necessity for such flexibility in this setting. In this model, four operational levels (National/Country, Organizational, Department, and Employee) were used to represent the TWH practices of a multinational organization and the possible areas of intervention to ultimately influence workforce safety, health and well-being outcomes (see Figure 1). Similar to many TWH frameworks, we draw upon the socioecological model to focus on the conditions of work that give rise to health [3]. Specifically, we adapt the multi-level model of safety described by Burke and Signal [28] to demonstrate the influence of social systems on workforce outcomes.

At the national level, each country is influenced by cultural values, political and economic structures, and available community resources. These characteristics may vary across countries and thereby generate differences in health and safety work practices within an international organization operating across culturally diverse locations. These national/country level characteristics in turn influence the organizational level, as cultural values shape senior leadership attitudes towards the development of TWH policies and programs and their implementation in practice as manifested by a culture of TWH in the entire organization. Especially in large organizations that operate across countries, there may be departmental differences in the implementation of TWH initiatives. As such, leadership and culture may vary by department. It is at the organizational and department levels where we can use TWH interventions to facilitate transactional changes in the organization via the implementation of TWH policies and programs. Furthermore, lasting transformational change can also be implemented at these levels through leadership development and promotion of effective transformational leadership practices [27,28,29,30]. Ultimately, our goal was to influence employee level outcomes through the adoption and implementation of organizational and department level TWH interventions with the aim to improve employees’ perceptions of the health and safety climates (i.e., perceptions that their organization/department is committed to their health and safety) as well as their knowledge of TWH.

In the present paper, we provide an example of how this conceptual model can be operationalized with a multinational agribusiness situated in Central America. This paper outlines our assessment and survey processes. In doing so, we aim to demonstrate what information should be collected to assess how the company is implementing the TWH approach at the department, organizational, and country levels as well as what information should be collected from employees to assess program impact. We then detail the trainings and recommendations that were informed by the assessments. Noting that, per the socioecological model, the trainings and recommendations focused on changing the organization through leadership training and program development, not on changing employee behavior. We conclude with a reflection and discussion of lessons learned for moving forward with TWH research and research-to-practice in international settings.

## 2. Case Study: A Model to Apply the TWH Approach to Latin America

### 2.1. Collaboration Background

Pantaleon is a multinational agribusiness headquartered in Guatemala with operations in four other countries within Latin America. In 2018, Pantaleon ranked 11th in the world for the production of sugar from sugar cane and was the second largest producer in Latin America, employing 13,994 people on average annually and exceeding 21,200 workers during the 6-month harvest season [31]. In 2016, prior to the start of the collaboration with the Center for Health, Work & Environment (CHWE) based at the Colorado School of Public Health, Pantaleon had already invested in efforts to improve their safety culture by hiring safety consultants and a corporate level environmental health and safety manager, as well as through adoption and continuous improvement of safety policies, practices, communications, and training programs. The company also had some existing health programs available to workers and their families including onsite health and dental clinics staffed by a team of occupational health nurses and physicians, chronic disease screening, health fairs, and health education programs. However, safety and health teams were not well integrated. Prior to the start of the collaboration, the safety program and health promotion program operated as separate entities in different departments of the company. The company sought to collaborate with CHWE to develop TWH approaches that expanded and integrated their established safety culture with an enhanced health culture.

In 2016, Pantaleon approached CHWE to learn how they could apply TWH principles to address the wide variety of health and safety hazards that commonly affect workers in the agriculture industry. Pantaleon and CHWE signed a memorandum of understanding with the main purpose of evaluating and improving the health, safety and well-being of Pantaleon’s diverse workforce using a TWH approach. As the collaboration evolved, CHWE researchers assisted the company with identifying priorities and in addressing several of them, such as Chronic Kidney Disease of unknown cause (CKDu), hearing conservation & noise mitigation in the sugar processing mills, worker nutrition, hydration, and injury and illness surveillance [32,33,34,35,36,37]. While progress was made in these specific areas, it was clear that the company could benefit from a comprehensive TWH needs assessment and leadership assessment to identify additional priorities for intervention.

For the purpose of this case study, we focus on our collaborative work to complete TWH needs and leadership assessments and the delivery of a leadership training program based on the results of these assessments as an effort to institute transformational change across all levels of the organization. CHWE adapted these activities from existing TWH research and practice activities conducted amongst small- and medium-sized businesses in the United States [27,38].

This case study was performed as a consulting project. The protocol was reviewed by the Colorado Multiple Institutional Review Board (COMIRB, #20-1158) and approved under category 4 exemption. In order to ensure confidentiality, all data analyses were performed by the team at CHWE. Pantaleon received only aggregated data summaries.

### 2.2. Assessment of Total Worker Health

Pantaleon, like other multinational agribusinesses, is a multifaceted and complex organization, which required a system-wide approach with the flexibility to meet a variety of TWH needs in multiple countries and departments. The focus for this study was on three of the six countries in which Pantaleon operates (Guatemala, Nicaragua, and Mexico), with the largest number of employees, including field workers, mill workers, and office workers. The organizational leadership structure of Pantaleon has a single corporate office directing operations in all three countries, within which department managers oversee department heads, who oversee the operational workforce (see Figure 2). The TWH assessment was conducted at each level of the organization, and included five main components: (1) key informant interviews and focus groups, (2) organizational assessment, (3) leadership self-assessment, (4) employee health and safety culture assessment, and (5) employee health risk assessment.

The purpose of the TWH assessment was to obtain a baseline estimate of the current state of TWH at the company as a whole as well as in each of its parts (different countries and departments). The results of the assessment were used to: (a) inform recommendations for new and changes to existing health and safety programs, policies and practices, and (b) assess leaders’ understanding of TWH and, in turn, develop and implement TWH leadership training for company management to help leaders utilize the results of the assessment.

CHWE investigators evaluated the current state of TWH policies, practices, and outcomes at each national branch in Guatemala, Nicaragua, and Mexico, as well as at the corporate headquarters in Guatemala City. The assessment was made up of the several components that are listed in Figure 3 and described below. The methods were based on a comprehensive approach we developed and used with small- and medium-sized businesses in the United States [27], which focuses on collecting information from all levels of the organization to obtain multiple perspectives. The following is an overview of our assessment strategy as adapted to this new setting. All assessments were administered between November 2018 and July 2019.

Assessment questionnaires were administered via multiple modalities, online using Qualtrics (Qualtrics, Provo, UT, USA), and in person using electronic tablets; details regarding each assessment are described below. For the development, coordination of survey administration and follow up, we worked closely with the communications and medical teams at each national branch. We acknowledged the cultural and organizational diversity between countries, and thus recognized the need to have input from our local collaborators in the design and development of each survey. All assessments were adapted to be culturally and linguistically appropriate and relevant to the populations being interviewed. For a complete timeline of the assessment process see Figure 3.

#### 2.2.1. Key Informant Interviews and Focus Groups

To begin the needs assessment, we conducted semi-structured focus groups and interviews with 171 key informants in three countries, including corporate managers, department managers, and department heads. The purpose was to gain a broad understanding of leadership’s perspective about the organization’s current efforts to promote and maintain the health, safety and well-being of all employees, as well as their own individual efforts and actions taken as leaders for employees in their respective areas of work. These activities allowed for a qualitative evaluation of engagement and knowledge, as well as to establish buy-in for TWH participation among leaders via leadership training (described below). Participants included corporate managers, their subordinate department managers, and department heads who oversee the field operations, which include administrative and operational workers across six main departments existing in all countries (agriculture, industrial, finance, human resources, logistics, and general management), plus six other departments unique to Guatemala (laboratory, occupational health and safety, civil engineering, business management, security, and information technology) and four departments unique to Mexico (agriculture management, mechanized harvest, AMAJAC, and TRE) (Figure 2). We conducted one hour-long in-person focus groups with department heads (n = 142), keeping each group to an average of 5 participants in order to facilitate a dynamic exchange of thoughts and perspectives as well as to be efficient with leaders’ time, interviews were conducted in Spanish. Next, we interviewed corporate and department managers (n = 29) in one-on-one 30-min-long meetings to allow for candid discussion of their perceptions regarding the state of TWH at the organization and their own leadership skills related to health and safety, interviews were conducted in Spanish and English depending on the leaders’ comfort. The same questions were used for both focus groups and interviews. The interviews and focus groups also gave the CHWE team the opportunity to explain the upcoming assessment process that would take place with their employees in order to gain buy-in and promote participation.

#### 2.2.2. Organizational Assessment

The organizational assessment was completed by the four lead human resource managers, including one at each of the three countries and at corporate headquarters (see Table 1). We adapted the organizational assessment from the Healthy Workplace Assessment^TM^ developed by Health Links™, a community-based TWH program focused on building the culture of health, safety and well-being in businesses of all sizes [38]. The Health Links assessment includes six benchmarks to measure the organization’s level of engagement around health and safety policies and programs: (1) organizational supports, (2) workplace assessments, (3) health policies and programs, (4) safety policies and programs, (5) engagement, and (6) evaluation (Table 1). The tool was developed by distilling constructs from the CDC Worksite Health Score Card [39], NIOSH TWH approach [1], and the WHO Healthy Workplace Framework [40].

We adapted the assessment from the Health Links™ version based on our knowledge and experience working with the organization during the prior 3 years. We added demographic questions targeted to the diversity of the working population including field and office workers, as well as seasonal workers and contractors. We also included questions aimed to measure the organization’s efforts to address specific health risks affecting the workers, such as consumption of alcohol, chronic disease, and infectious diseases. Finally, the workplace assessment benchmark, which measures employee surveys and employee health risks factors, was changed to be unscored given that the organization was just beginning to address this benchmark through the administration of these assessments. The version adapted for the current case study was an online assessment, administered via Qualtrics and consisted of 67 questions that took approximately 30–40 min to complete. Once the survey was completed and analyzed, a report card containing the results was generated and shared with each of the human resource managers. The report card was also later used in the leadership training, described below.

#### 2.2.3. Leadership Self-Assessment

Corporate managers, department managers, and department heads in each country completed a leadership self-assessment. This survey was adapted from three main sources (1) the Health Links Healthy Workplace Assessment as just described, (2) the Small+Safe+Well (SSWell) study, which aims to measure organizational change through the implementation of organizational-level TWH practices [27], and (3) the qualitative information gathered during the in-person key informant interviews and focus groups, where we identified the need for more detailed questions from the leadership (i.e., number of workers in their department, and worker demographics). The aim of this assessment was to measure leaders’ commitment to and promotion of health, safety and well-being in their respective departments. The online assessment administered in Spanish via Qualtrics was 72 questions and took approximately 25–35 min for leaders to complete. The assessment was distributed among all leaders in the organization including corporate (N = 40) and department managers and department heads in all locations (Guatemala = 53, Nicaragua = 38, Mexico = 36). A total of 167 leaders (response rate: 84%) completed this survey. See Table 1 for a complete description of survey constructs.

#### 2.2.4. Employee Health and Safety Culture Assessment

Administrative employees, including supervisors, coordinators and administrative assistants, and operations staff completed an employee health and safety culture assessment (Figure 2). We adapted this assessment from the SSWell study, as described above [27]. The survey sought to assess the culture of health, safety and well-being at the organization from the employee’s perspective. Survey questions focused on both health and safety measures (Table 1).

Due to the large size of the workforce totaling over 17,200 during the 2018–19 harvest season, we took a two-pronged approach to administer this assessment. The first format was an online survey for office-based work units, which was administered through Qualtrics. The online survey was sent out to all administrative employees with access to a computer in all three countries. This survey was 120 questions long and took 30–40 min to complete. A total of 346 employees participated in the assessment (response rate: 40%). The second format was administered in-person to operational workers by local interviewers who we trained to record data on portable tablets. This second format was used in order to accommodate the range of education and literacy levels among operational employees without computer access (e.g., field and industrial mill workers). The CHWE team worked together with the organization’s local medical teams to simplify the in-person survey questions while still incorporating all of the assessment constructs. Once translated to Spanish, the questions were reviewed by the medical teams at each site, who are in close daily contact with the target populations. Based on their input, modifications included visual analog scales to administer all Likert scale questions. Finally, we conducted a pilot of the assessment with a small group of field workers, who pointed out additional adjustments needed to account for language, cultural and educational differences. The final in-person culture survey was 70 questions long and took 25–35 min to complete. Surveys were conducted on a 20% convenience sample from each department, except the largest department, Agriculture, from whom we collected a convenience sample of 297 workers (3% of the 9744 Agriculture Department workers), bringing the total in-person surveys conducted to 1541.

#### 2.2.5. Employee Health Risk Assessment and Evaluation of Clinical Consult Data

Furthermore, as part of the employee health and safety culture assessment, employees also completed a brief health risk assessment (HRA) questionnaire adapted from the SSWell study [27]. Questions pertained to perceived overall health and well-being, physical activity, stress, behaviors including tobacco and alcohol use, fatigue, absenteeism and presenteeism (Table 1). The combined data from the HRA and clinical consults provided the opportunity to evaluate the most critical health outcomes affecting the workforce at each site. Finally, because we were working with a large, self-insured company, we were able to obtain aggregate data on employee clinic visits from the three national branches (Guatemala, Nicaragua and Mexico). De-identified data collected included reasons for medical visits and diagnoses from years 2017–2019.

### 2.3. Total Worker Health Leadership Training

Understanding that leadership and climate significantly contribute to the TWH culture of the organization [41], we aimed to leverage the role of leaders to stimulate a sustainable organizational culture transformation. Informed by the results of the assessments, CHWE designed a TWH leadership training specifically tailored to Pantaleon leaders at each national branch including corporate and department managers, and department heads. Trainings were delivered from September through December 2019 following the completion of the assessments (Figure 3). Trainings followed the same format for all national branches. Participating leaders in each country received educational packets, which included their assessment results pertinent to their country and department, along with a goal setting activity sheet that was used throughout the training as described below. The class size was limited to 20 leaders from all departments per training and lasted 4 h. A total of 120 leaders participated in the training (Guatemala = 51, Nicaragua = 27, Mexico = 31, Corporate = 11).

The purpose of the trainings was to help company leaders learn and incorporate TWH leadership strategies into their business practices in order to enhance worker health, safety and well-being [41]. The content of the training incorporated the results of the assessments and was based in large part on the training developed by the SSWell study [27]. As in the SSWell training, we focused on validated leadership theories and core TWH principles to help address the leader’s own health and that of their employees, with the ultimate goal of improving TWH climate of the organization and departments.

To optimize applicability to this industry and employer, we made five fundamental modifications in our training adapted from the SSWell study. First, the SSWell training provides leaders with their results aggregated at the company level, allowing leaders to reflect on their own self-reported TWH leadership skills, employee’s perspective of a healthy and safe workplace, and the current TWH policies and programs of the organization. We expanded upon this approach by addressing the unique cultural differences between countries, departments, and leaders in our presentation of the assessment results from the leaders’ own country and department, with the purpose of providing leaders with personalized tools to aid in their decision making and prioritize action [42]. For example, we included summarized results (e.g., health and safety motivation, organizational supports, leadership commitment, levels of self-reported stress) from the employee health and safety culture assessment by department in the training materials. Additionally, based on the most common health risks identified in the health risk assessment results, we focused on three main health promotion priority areas in the training: (1) chronic disease prevention, (2) sleep hygiene, and (3) preventive stress management and mental health. Priorities were identified to meet the needs across our conceptual model of organizational, leader, and employee risk and benefits (Figure 1) weighing the potential risk and severity of each health outcome with (a) the potential for the company to be able to help prevent that outcome, (b) the potential benefit to worker health and safety and (c) the potential benefit to company productivity by reducing the risk.

Notably, while we did include information related to occupational safety, it was not heavily emphasized, except to reinforce concepts that leaders had learned in safety culture trainings the prior year. Second, because leaders came from the same organization, we were able to facilitate a discussion about the similarities and differences in TWH within and between countries and departments, to encourage cross-pollination of ideas. Third, the team delivering the training was comprised of three English speakers and one native Spanish speaker. We hired bilingual facilitators and interpreters from each country to help us deliver the trainings in a manner that was culturally representative for each country. The English speakers gave the main didactic portions, while the native Spanish speaker facilitated the discussions (e.g., goal setting discussions). Fourth, due to the high work demands at the operations, we adapted the original 8-h training to 4 h to meet the schedule needs of the leaders. Finally, we used results from the assessment as prompts to facilitate group discussion in order to generate a list of suggestions of ways in which the corporation could better support leaders and their employees in adopting TWH best practices, as described below.

#### 2.3.1. Total Worker Health Leadership Training Components

The training was divided into four modules (see Table 2 for a detailed description). The first module focused on the fundamentals of TWH and how it applies to the role of organizational leaders. The second focused on TWH leadership best practices, including personal well-being and mental health as well as impacts of fatigue, sleep and chronic conditions. At the end of the second module, leaders were asked to set a realistic and attainable personal goal to improve their own safety, health and well-being. The third module focused on employees’ perspective of a healthy and safety, relying on the employee culture assessment results. At the end of this module, leaders were asked to set an attainable goal on something they could do as leaders to promote the health and well-being of the workers in their departments. The fourth module focused on TWH and organizational sustainability through a review of each country’s organizational TWH policies and programs. Because ultimate decision-making authority resides at a higher organizational stratum, we did not invite participants to set overall corporate goals.

As a closing brainstorming exercise, leaders were asked to rank their organization as a whole, using the six benchmarks addressed in the organizational assessment. Leaders were then presented with the actual results of the report card from their own national branch. Discrepancies were used as starting points for open-ended conversation about the organization’s efforts promoting health and safety and areas for improvement. Leaders were then invited to provide feedback and suggestions to the organization about how to improve the culture of health, safety and well-being. These suggestions were distilled into major themes and were presented back to the organization’s corporate management in the form of program and policy recommendations.

#### 2.3.2. Corporate Recommendations

In addition to the leadership trainings, a key outcome of the company-wide TWH assessment was the development of organizational program and policy recommendations related to employee health and well-being. The CHWE team analyzed the results of the interviews, survey assessments and clinic visit data in order to identify opportunities and priority areas of focus for the company as a whole and for each country site. This analysis helped identify priority areas for corporate investment in order to have maximal impact. Priority areas identified by the CHWE team and Pantaleon leaders were: (1) maintain the established safety culture in the organization, (2) improve employee health and well-being through the development of a sustainable health and wellness culture that mirrors the thoroughness and commitment of the established safety culture, (3) chronic disease prevention, (4) preventive stress management and mental health, and (5) sleep hygiene and fatigue. Table 3 outlines the recommendations provided to the organization that were informed by the data collected in the assessments (see Section 2.3 above for priority selection). The CHWE team developed and presented to leadership several evidence-based programs and policy recommendations to help address each priority area. These included protocols for implementation and evaluation that could then be adapted by the company at each country site. Country specific recommendations were informed by the brainstorming group activity held during the training, (see Table 2). Final recommendations were provided in a series of presentations made to leadership including the executive team, human resources, the health and safety teams, as well as the company’s advisory board on responsible development.

## 3. Discussion (Lessons Learned)

This case study illustrates how to apply a TWH approach to a large, multinational agribusiness in Latin America. Through this process, we learned that with a close private-public collaboration, conscious efforts to obtain buy-in at all levels of the organization, and attention to appropriate cultural and linguistic differences, a TWH model that we originally developed for businesses in the United States can be applied to a large agribusiness operating in multiple Latin American countries. Core concepts around leadership, culture, health and safety have cross-cultural relevance. However, we learned that the adoption and implementation of such concepts will vary by country and department. It is important to adapt and evaluate the implementation of the TWH approach in Latin America as the workforce represents a diverse cultural setting with a potentially vulnerable worker population that has received little attention in the literature.

We propose that our conceptual model in Figure 1 has broad applicability across borders, industries, and size of business. In our application of it with a multinational agribusiness in Central America, we made sure to complete the TWH assessments at the corporate, department, and employee levels. As we discuss above, it required a systematic strategy to capture information from all levels depicted in Figure 2. The combined findings then informed our leadership training strategy to train leaders from all company levels in TWH leadership. This approach produced rich information about how a large, multinational agribusinesses can implement a TWH approach and a mechanism to communicate this information across all levels of company leadership. The importance of TWH conceptual models have been stressed, but their use is seldom demonstrated in research and practice [3]. With this case study and our prior work applying it to small business in the United States [27], we demonstrate that the model has broad applicability. We were successful in applying the conceptual model to this setting, but it was more challenging than when we have applied it to individual small businesses. The large company described in this case study operates across multiple countries and departments, making it challenging to devise assessment and training strategies with broad reach and uniform applicability. We found that we needed to conceptualize each department within the large company as a “small business.” While all units operate under an overarching corporate structure and have certain shared values, resources, and business objectives, we observed that each operational unit has local safe and health culture, much as we see when working with independent small businesses.

It is important to note that because the company had already spent several years incorporating safety culture constructs into their business model, including both a safety management system and previous safety leadership training, we were able to focus on the gap in health promotion. As such, our TWH assessment process largely focused on health and well-being and found particular needs in the areas of mental health, sleep, and chronic health conditions. Had we needed to introduce more content on safety into the leadership training, the process would have required a reprioritization, more training time, additional training modules, and additional goal setting. We are currently examining the corporation’s safety data as part of a parallel research initiative [32].

The timeframe in which we were able to conduct the work described above may be deceiving. The academic team at CHWE and the partners at Pantaleon were able to build upon a pre-existing two-year working relationship that had already applied TWH principles to the specific problem of chronic kidney disease in the workforce [33], and to collaborate in enhancing their hearing conservation program [37]. The level of trust and collaboration allowed us to successfully collect data across many levels in the organization, while building and gaining more buy-in and participation, and at the same time ensuring protection of workers’ personal health and survey data.

As more work is conducted in culturally diverse settings, it is important for occupational health and safety researchers and professionals to gain some cultural training [43]. By showing an interest in the culture and maintaining cultural flexibility we were able to successfully navigate the culture’s intricacies. Our ongoing collaboration with the organization has allowed us to gain a deeper understanding of the employees’ perceptions and approaches to health and safety in the organization. This allowed us to more effectively work cross-culturally in this multinational organization, understanding our own cultural biases and how the culture at each national branch shapes the organizational climate. In this case study each country was treated separately, but with the overarching consent to intervene coming from the central corporate headquarters, permitting a unified approach while still allowing for unique cultural differences between countries, departments, and leaders, influenced in part by each country’s political and economic structure.

In addition to examining the impact of TWH leadership training on the business’s health and safety climate, future research with this workforce will enable us to test hypotheses related to the international TWH conceptual model (Figure 1). For example, we can examine the underlying work-related determinants of the measured levels of stress and fatigue from the employee health and culture assessment and evaluate the relationship with other health, safety and productivity outcomes. We will be relying on the “RE-AIM” public health evaluation framework [44], in particular, which can be used to both design interventions as well as evaluate them for reach, efficacy, adoption, implementation, and maintenance (RE-AIM), including our training and the TWH policies, programs and practices that derived from this process, as described above.

## 4. Limitations (Challenges)

It is important to acknowledge that this work represents one of the few documented examples of the TWH approach in Latin America, and in the agricultural industry generally. However, this case study is only representative of a single agribusiness in Latin America, and so the dissemination and generalizability of this approach to other industries, regions, and businesses will require further study. While participation in the assessments was promoted by leadership, it was voluntary, therefore selection bias was possible. For participants of the in-person employee health and safety culture assessment, we collected information from a convenience sample during work hours. We addressed this limitation by stratifying our sampling by worker groups and aimed to reach a representative 20% sample of each.

## 5. Conclusions

As the global economy grows and diversifies, expanding TWH to more diverse audiences will become increasingly important. We present the methodology of this case study because it suggests the applicability of TWH in Latin America and demonstrates the feasibility of applying TWH concepts in agribusiness and in cross-cultural settings. We demonstrate that this approach can be used to identify priority areas for investment and action in TWH. In the long term, this approach is likely to have a measurable impact on employee health, safety and well-being, and ultimately productivity. In future research, we aim to examine the impact of the TWH leadership training on health and safety climates as well as workforce health and productivity outcomes. We also aim to examine relationships described in our conceptual model (Figure 1) using data collected during this case study.

## Figures and Tables

**Figure 1 ijerph-18-02252-f001:**
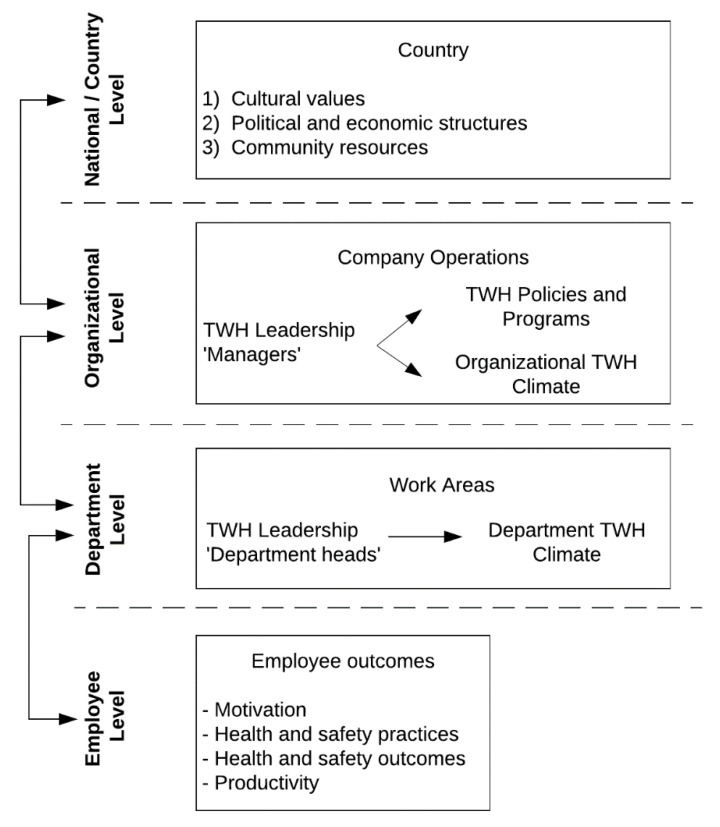
International Total Worker Health (TWH) Conceptual Model.

**Figure 2 ijerph-18-02252-f002:**
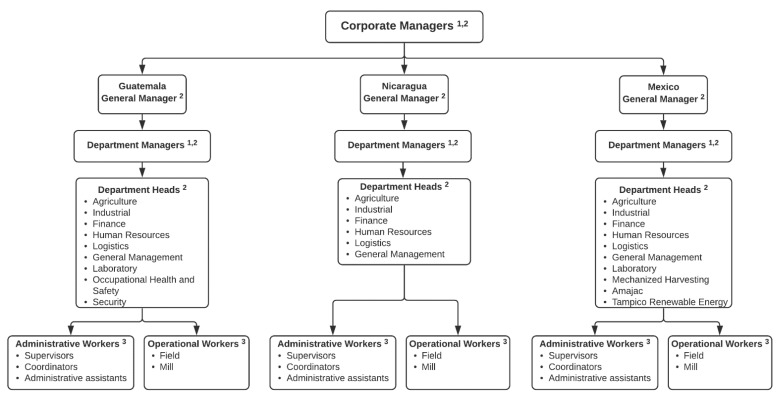
Pantaleon Organizational Chart. The needs assessment was conducted at all levels of the organization, including headquarters and all departments within each national branch. Footnotes indicate employees who participated in (1) the organizational assessment (human resource managers only), (2) key informant interview and focus groups, and leadership self-assessment, and (3) the employee health and safety culture survey and the health risk assessment.

**Figure 3 ijerph-18-02252-f003:**
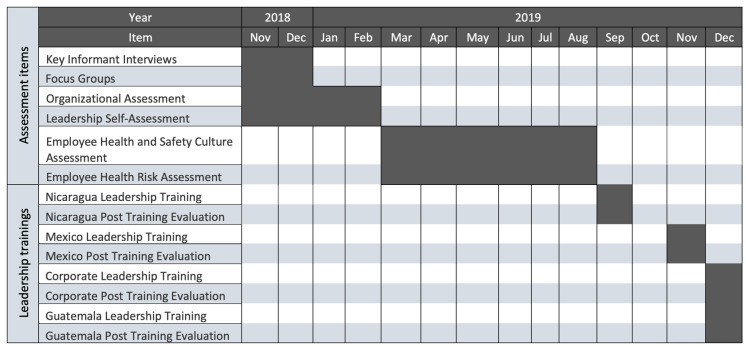
Timeline of TWH Needs Assessment and Leadership Trainings at Pantaleon.

**Table 1 ijerph-18-02252-t001:** TWH Assessment components and their constructs.

Assessment Components				
**Assessment Constructs**	**Organizational Assessment**	**TWH Leadership Self-Assessment**	**Employee Health and Safety Culture**	**Employee Health Risk Assessment**
	Organizational Supports	Leadership Supports	Leadership Supports	Employee well-being
	Health Policies and Programs	Employee Workplace Assessment	Organizational supports	Overall health
	Safety Policies and Programs	Support for Health Programs and Policies	Health and Safety motivations	Work stress
	Engagement	Support for Safety Programs and Policies	Organizational commitment to health and safety	Tobacco use
	Evaluation	Leadership Engagement	Health and safety climate	Physical activity
		Health and Safety Program Evaluation	Health and safety behaviors	Absenteeism
		Leadership Practices	Psychological safety	Presenteeism
		Department Demographics	Leadership commitment to health and safety	Fatigue
			Access to benefits	
			Perceived work hazards	

**Table 2 ijerph-18-02252-t002:** International Total Worker Health Training objectives and description.

Total Worker Health Leadership in-Person Training Objectives and Description—4-h One-Day Training
(1) Increase leaders’ awareness of the business case for worker health and well-being a. Definition of TWH b. The role of the organization in creating and promoting a TWH culture c. Understanding your role as a leader in creating and promoting a TWH culture d. The 5 practices of successful leaders in creating culture of well-being
(2) Educate leaders about their own health and well-being needs a. The importance of leaders’ own personal health in order to be effective leaders b. Impacts of stress, sleep, and physical activity. What can leaders do about it? c. Needs assessment review: Review of results from leadership assessment with other leaders d. Activity: Describe your own health and well-being and identify which areas you want to work on as a leader to demonstrate the 5 practices of successful leaders in creating a culture of well-being e. Goal setting: Generate a specific, actionable goal for contributing to your personal health and well-being
(3) Educate leaders about their own employees’ health and well-being needs a. Identifying employees’ value(s) for their health based on survey results b. Review of employee’s Culture Survey results: views of company and views of leadership c. Needs assessment review: Review results from employee health and safety culture assessment from each department d. Activity: Describe your department’s culture and identify which aspect of your department’s culture you want to work on as a leader to demonstrate the 5 practices of successful leaders in creating a culture of well-being e. Goal setting: Generate a specific, actionable goal for contributing to your employees’ health and well-being
(4) Educate leaders about TWH from the organizational level and how the organization can improve well-being of the workforce a. Review of the benchmarks for TWH success b. Needs assessment review: Review of results from Organizational assessment (benchmarks results) c. Brainstorming group activity: Ways the organization can promote worker health & well-being at the Organizational level

**Table 3 ijerph-18-02252-t003:** Recommendations for the implementation of a TWH approach at Pantaleon.

Outline of TWH Recommendations
(1) Maintain established workplace safety programs and policies
(2) Integrate health and safety programs through the inclusion of wellness and wellbeing programs
(3) Address the prevention of chronic diseases a. Chronic non-occupational diseases (e.g., Diabetes, hypertension) b. Physical inactivity c. Alcohol and tobacco use prevention d. Nutrition
(4) Address stress and mental health a. Occupational stress prevention training b. Work-life balance programs
(5) Address sleep deprivation and fatigue a. Sleep hygiene programs b. Administrative changes to reduce fatigue c. Work schedule revision and improvement

## Data Availability

Not applicable.

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
