# Peer review of "International Total Worker Health: Applicability to Agribusiness in Latin America"

_ijerph, 2021, doi:10.3390/ijerph18052252_

Round 1
Reviewer 1 Report
Results and conclusions need a deeper analysis in order to justify the research aims.

Author Response
This manuscript aimed at presenting a case study about the application of TWH approach to the Latin American agriculture industry conducted from 2018 to 2020. This case study explored three novel concepts: the applicability of TWH theories in a Latin American cultural setting, the applicability of TWH in a vertically integrated agribusiness, and the relevance and applicability of the TWH approach within a multinational organization.
I strongly recommend reconsidering the manuscript after making minor revisions. The acceptance of manuscript would depend on making special revision:
The question is original and it is well defined. The aim of the study is well defined. But the introduction is too short and it lacks an analytical report of way the conceptual model was conducted.
We have included more background on the theories that informed our conceptual model as well as information on how we applied the model in our case study (see lines 90-93, 115-125).
The results are well defined but they are not analytical so they cannot explain clearly the study’s aim.
The reviewer raises an important point. To improve the clarity of the manuscript, we have decided to restructure our manuscripts sections. We no longer have a “Methods” and “Results” section as our manuscript does not present such information. In our manuscript, we focused on the methodological and conceptual approach of how to conduct international TWH research. We have made this clearer at the outset and in the conclusion (see lines 126-127, 430, 502, 513, 115-125, 515-517)
The conclusion is too short. It needs a deeper analysis in order to justify case study’s aim.
We believe we have provided a thoughtful commentary on our lessons learned and the conclusions we can draw from our experience. Given our focus on methodology, we chose to expand the discussion to include a reflection of the use of our conceptual model in this population as it compares to its use in other settings in the United States. (See lines 441-462)
The manuscript lacks limitations
Thank you for your comment. We have now included a section describing the challenges faced though the implementation of this project, which we believe make for a stronger manuscript (Lines 502-512)
The quality of presentation had met many standards of presenting.
The manuscript has an interest for the readers. It is an original analysis of a TWH approach that, it might under minor revisions to have an international response.
TWH approach is a core issue in reducing inequality in workplace. So, a manuscript describing of how TWH approach can be applied to a multinational agribusiness support the development of a healthy and safety workplace might have an overall merit. It increases the knowledge in leadership, health and safety cross – cultural relevance.
Reviewer 2 Report
The study is of high quality and interest. I have only a few comments.
Lines 113, 114, 115 contain three titles (title, subtitle, sub-subtitle). Not only does it seem strange but it is also unclear.
The company Pantaleon is introduced too broadly, whereas the TWH concept is presented briefly and quite late in the paper.
My main concern is methodological and should be addressed in the article. There is stated in chapter 2.2.2 that lead human resource managers completed the organizational assessment (line 231-232). How is ensured that the assessment is really objective and relevant? Would it not be more appropriate to have external evaluators?
Author Response
The study is of high quality and interest.
I have only a few comments. Lines 113, 114, 115 contain three titles (title, subtitle, sub-subtitle). Not only does it seem strange but it is also unclear.
The reviewer raises an important point. To improve the clarity of the manuscript, we have decided to restructure our manuscripts sections. We no longer have a “Methods” and “Results” section as our manuscript does not present such information. In our manuscript, we focused on the methodological and conceptual approach of how to conduct international TWH research. We have made this clearer at the outset and in the conclusion (see lines 126-127, 430, 502, 513, 115-125, 515-517)
The company Pantaleon is introduced too broadly, whereas the TWH concept is presented briefly and quite late in the paper.
TWH is introduced in lines 35-112, and we believe it is detailed enough for readers to establish its applicability. However, we defer to the editor and can provide more detail if the editor wishes us to.
My main concern is methodological and should be addressed in the article. There is stated in chapter 2.2.2 that lead human resource managers completed the organizational assessment (line 231-232). How is ensured that the assessment is really objective and relevant? Would it not be more appropriate to have external evaluators?
Thank you for your comment. We agree that ideally, we would have external evaluators. For the organizational assessment we used a framework previously tested and used in small business in Colorado, USA (Tenney et al. 2019); this study found that the person completing the assessment did not bias the results. Furthermore, during our leadership training we had the opportunity to internally validate the HR results when presented and discussed by the leaders participating in the training. (Lines 395-403)
Tenney, Liliana, et al. "Health links™ assessment of total worker health® practices as indicators of organizational behavior in small business." Journal of occupational and environmental medicine 61.8 (2019): 623.
Reviewer 3 Report
Total Worker Health® (TWH) is a framework integrating the safety, health and well-being of employees and workplaces that has been successful in Europe and the USA. This framework has not been implemented in Latin America or the agricultural sectors, leaving large and vulnerable populations under-represented in implementing and evaluating these strategies to improve safety and promote health and well-being.
The paper presents the adaptation process and strategy for conducting TWH assessment at many organizational levels and in many countries, as well as the recommendations and feasibility of conducting TWH in Latin America.
The article presents an interesting conceptual model to be implemented, but I suggest that:
- improve Figure 2 and Table 3 as it is not very legible
- recommendations, perhaps consider a graphic form, or partly a graphic form
- expand the conclusions
Author Response
Total Worker Health® (TWH) is a framework integrating the safety, health and well-being of employees and workplaces that has been successful in Europe and the USA. This framework has not been implemented in Latin America or the agricultural sectors, leaving large and vulnerable populations under-represented in implementing and evaluating these strategies to improve safety and promote health and well-being.
The paper presents the adaptation process and strategy for conducting TWH assessment at many organizational levels and in many countries, as well as the recommendations and feasibility of conducting TWH in Latin America.
The article presents an interesting conceptual model to be implemented, but I suggest that:
Improve Figure 2 and Table 3 as it is not very legible
Thank you for the suggestions on our figures and visual interpretation of this work. We have updated both figures to make them more legible and easier to interpret (Lines 183, and 199)
recommendations, perhaps consider a graphic form, or partly a graphic form
We are including more details on the recommendations to the organization, as well as including a table outlining recommendations in more detail (Lines 413-420, 428-429)
expand the conclusions
We believe we have provided a thoughtful commentary on our lessons learned and the conclusions we can draw from our experience. Given our focus on methodology, we chose to expand the discussion to include a reflection of the use of our conceptual model in this population as it compares to its use in other settings in the United States. (See lines 441-462)
Reviewer 4 Report
Dear Authors
I carefully evaluated your paper, founding it overall well written, but not so well presented. The paper focuses on a relevant issue that needs to be properly investigated. The scientific background is well established and the implication are also relevant.
Nevertheless, the main concern about this paper regards the results section. In particular, the results of the assessment of all your variable of interest are not clearly presented. I expected some tables of results, but I can not find any in your results section. I suggest to strongly improve results section, in order to clearly present what results from the assessment strategy, and then comment them in the discussion section. It is also unusual to add references in the results section.
You mixed results and discussion, but they should be clearly separated. Also it is not clear what is the analytical approach you used, in order to provide some relevant information on the data you have collected.
Best regards
Author Response
Dear Authors
I carefully evaluated your paper, founding it overall well written, but not so well presented. The paper focuses on a relevant issue that needs to be properly investigated. The scientific background is well established and the implication are also relevant.
Nevertheless, the main concern about this paper regards the results section. In particular, the results of the assessment of all your variable of interest are not clearly presented. I expected some tables of results, but I cannot find any in your results section. I suggest to strongly improve results section, in order to clearly present what results from the assessment strategy, and then comment them in the discussion section. It is also unusual to add references in the results section.
You mixed results and discussion, but they should be clearly separated. Also it is not clear what is the analytical approach you used, in order to provide some relevant information on the data you have collected.
We apologize for the confusion. To improve the clarity of the manuscript, we have decided to restructure our manuscripts sections. We no longer have a “Methods” and “Results” section as our manuscript does not present such information. In our manuscript, we focused on the methodological and conceptual approach of how to conduct international TWH research. We have made this clearer at the outset and in the conclusion (see lines 126-127, 430, 502, 513, 115-125, 515-517)
Round 2
Reviewer 4 Report
Dear Authors
Your manuscript is now more clear and well defined with respect to te first version. Well done.
Best regards